# Effect of Aeration on Blockage Regularity and Microbial Diversity of Blockage Substance in Drip Irrigation Emitter

Peng Li [1], Hao Li [1],*, Jinshan Li [1], Xiuqiao Huang [1], Yang Liu [1] and Yue Jiang [2]

[1] Farmland Irrigation Research Institute, Chinese Academy of Agricultural Sciences, Xinxiang 453002, China
[2] Research Center of Fluid Machinery Engineering and Technology, Jiangsu University, Zhenjiang 212013, China
* Correspondence: lihao01@caas.cn

**Abstract:** Aerated drip irrigation is rendered as a new water-saving irrigation method based on drip irrigation technology, which is endowed with the function of effectively alleviating the problem of rhizosphere hypoxia in crop soil, enhancing the utilization rate of water and fertilizer; as a result, it improves the harvest and quality of crops. However, clogged emitters are important indexes, among others, that pose an influence to the service effect and life duration of drip irrigation systems. At present, the working principle and mechanism of the influence of air feeding on the blockage of drip irrigation emitters remain unclear. Therefore, based on the two gas filling methods of the micro/nano bubble generator and Venturi injector, the dynamic change process for the average flow ratio of an air-filled drip irrigation emitter was studied by the method of emitter plugging test. 16S rRNA sequencing was used to analyze the microbial diversity of the emitter plugs. The results show that the air injection can pose influence on the clogging procedure of drip irrigation emitters, and more importantly, it makes the distribution of blocked emitters more uniform, thus improving the uniformity of the system. Different filling methods have different effects on the blockage of the emitter. Among them, the blockage time of drip irrigation system under the micro/nano aerated drip irrigation (MAI) mode is 5.73 times longer than that under unaerated drip irrigation (UVI), and similarly, Venturi gas drip irrigation (VAI) is close to that under UVI. The filling method changed the microbial diversity of the blockage in the emitter. Among them, the number of operational taxonomic unit (OTU) unique to MAI was 2.1 times that of UVI, and the number of OTU unique to VAI was 1.3 times that of UVI. Meanwhile, gas addition will inhibit the growth of Nitrospirae and Proteobacteria microbial communities and promote the growth of Firmicutes and Actinobacteria microbial communities. Furthermore, the increase in microbial extracellular polymer in the plugging material of the emitter was inhibited and the plugging process of the emitter was slowed down. The research results are of great significance in the disclosure of the clogging mechanism of drip irrigation emitter and constructing the green, anti-blockage technology of aerated drip irrigation.

**Keywords:** micro/nano aerated; venturi gas; blockage of the emitter; average flow ratio

## 1. Introduction

Aerated drip irrigation, as a new water-saving irrigation method, aims to provide the optimal ecological environment for crops, which can be adopted to realize the synchronous deliver of water and air required by crops and improve the problem of soil hypoxia [1,2]. It has been demonstrated by the studies that aerated drip irrigation can be capable of improving soil permeability in an effective manner [3–5] and increase crop water and nutrient utilization rates [6,7], which is endowed with great merits in technology over others with good prospects for application [8,9]. Many scholars have studied the influence of aeration on drip irrigation systems and found that aeration can effectively change the hydraulic characteristics of water flow, promote microbial metabolism in drip irrigation systems, and reduce the frictional resistance between water bodies and walls [10].

At the same time, aeration in irrigation water affects the delivery of particles by water flow, and changes the water density and viscosity coefficient [11,12]. As a major component of drip irrigation systems, the principal function of the emitter lies in the capability of enabling the pressurized water to flow through its internal narrow channel or micro hole to fully dissipate energy, and then drop into the soil with a uniform and stable flow rate. In order to realize its function, the flow channel or micro hole inside the emitter is generally only 0.5–1.2 mm [13]. In the process of irrigation, the suspended particles, chemical precipitation, microbial metabolites and other impurities in the water are subject to the blockage of the emitter, thus reducing the uniform distribution of the drip irrigation system. Consequently, the service result and duration of the whole drip irrigation system are affected in an explicit manner [14,15].

For a long time, many scholars have carried out a large number of studies [16–20] to solve or alleviate the blockage of drip irrigation nozzles by configuring filtration equipment, optimizing the structure of the drip nozzle flow channel, regularly flushing the nozzle and adding chlorine to irrigation water and other methods, with certain achievements. However, the clogging problem of the nozzle serves as the significant indexes among others restricting further development of drip irrigation technology. Based on the research of Li et al. (2020), the use of aerated irrigation can delay the blockage of the emitter [21], which offers a new solution to the problem of the blockage of the emitter. However, that mechanism of the influence of aerated irrigation on the blockage of the emitter is still unclear and needs to be further studied.

It is generally believed that emitter blockage is mainly manifested in three forms: physical, chemical and biological blockage [22,23], respectively, among which biological blockage is often the initiating factor of other blockages with great complexity and importance [13,23,24]. The larger the number of microorganisms in the irrigation system, the more complex the community, and the more likely the emitter is to suffer from biological blockage [25]. Aeration of irrigation water in the working process of aerated drip irrigation will inevitably change the microbial environment of the mentioned drip irrigation system, affect the structure of the microbial community and ultimately affect the process of irrigation blockage.

To have further exploration on the influence of aeration on blockage regularity and uniformity of drip irrigation systems, as well as the microbial diversity of the blockage material in the emitter, this paper takes aerated drip irrigation as the research object, and adopts two classical aeration methods, micro/nano and Venturi, to carry out the following research: (1) monitor the dynamic change process of the average flow ratio (Dra) of drip irrigation system under different feeding modes-,with studies on the clogging rule and also consider using distribution uniformity/uniform distribution of drip irrigation system under different feeding modes; (2) analysis on the microbial diversity of the clogging materials in the emitter based on varied filling approaches, and exploration on the microbiological mechanism of clogging in drip irrigation emitter. The research results are of great significance for revealing the blockage mechanism of drip irrigation emitter and constructing the anti-clogging technology of aerated drip irrigation.

## 2. Materials and Methods

### 2.1. Selection of Emitter

To study the effect of air filling on blockage of the emitter and microbial diversity of the Blockage substances, the commonly used inset patch type drip irrigation system in the market was selected in the experiment as the research object. Before the test, the flow-pressure relationship and the coefficient of variation ($C_v$) of the emitter were given verification as (ISO,9261) [26]. Among them, the relevance between flow rate and pressure of emitter is expressed in the following formula:

$$q_e = kp^m$$

In the equation, $q_e$: Emitter flow, L h$^{-1}$;

P: working pressure, MPa;

K denotes flow coefficient;

M refers to the flow state index.

The coefficient of variation ($C_v$, %) was calculated as follows:

$$C_v = 100 \times S_q / \bar{q}$$

In the equation: $s_q$: standard deviation of the flow rate of the emitter, L h$^{-1}$;

$\bar{q}$: average flow rate of emitter L h$^{-1}$;

After testing, the structure parameters and performance indexes of the emitter are demonstrated in Table 1. As can be seen from Table 1, the flow pattern index M of the emitter is 0.48. According to the quality standard of the emitter (ASAE Standards, 2003B) [27], the performance of the embedded patch type emitter is "excellent".

**Table 1.** Test emitter runner size and performance index.

| Emitter | Rated Flow (L h$^{-1}$) | Connection Method | Compensation Function | Flow Channel Size (mm) | | | Filtering Area (mm$^2$) | Discharge Coefficient $k$ | Flow Index $m$ | $C_v$ (%) |
|---|---|---|---|---|---|---|---|---|---|---|
| | | | | Width | Depth | Length | | | | |
|  | 1.0 | Set Within patch | No pressure compensation | 0.6 | 0.74 | 65 | 45 | 0.347 | 0.48 | 1.74 |

### 2.2. Test Device

The test was conducted at the Quality Testing Center of Water Conservation Irrigation Equipment of the Ministry of Water Resources. The test was configured as three experimental groups: UAI (unaerated drip irrigation), VAI (Venturi gas drip irrigation), MAI (micro/nano aerated drip irrigation). The test device and its principle are shown in Figure 1. The test device is composed of a head and a test part. The first part includes four pools (single volume 0.5 m$^3$), a micro and nano bubble generator (Yunnan Summer spring, Kunming City, Yunnan Province, China), a Venturi ejector (Mazzei injector corp, Bakersfield, CA, USA), a frequency conversion pump (rated flow rate 3 m$^3$ h$^{-1}$, head 50 m, power 1.5 kW), a filter (120 mesh), a valve and a pressure gauge (0.4 level). The test part includes 3 groups of test beds, each of which is installed with 5 drip irrigation belts (1 drip irrigation belt of E1–E5, 10 m in length and 0.2 m in spacing). The drip irrigation belts are installed on the test bench, and the bottom of the test bench is a sink with a certain slope, and the sink is connected with a water tank, which can be used for the collection of test medium through the emitter into a water tank.

The Venturi ejector was installed at the front of the Venturi gas drip irrigation (VAI), and the micro/nano bubble generator was installed at the front of the micro/nano aerated drip irrigation (MAI), with the gas rate of 3 L min$^{-1}$. The unaerated experimental group (UVI) was not aerated, and neither Venturi ejector nor micro/nano bubble generator was installed at the front, and the other devices were consistent with the aerated experimental group. To further eliminate the effect of physical blockage on the test results, a circulating water supply system was adopted for this test, where the tested water source was accessed from a reservoir after settling in the front of the reservoir and then entered the test pipeline. After repeated precipitation and filtration, the contents of suspended particles and solid impurities in the water were cut down, and influence of physical blockage on the experiment was correspondingly reduced.

In order to monitor the clogging process of the emitter in each drip irrigation system, the emitter flow rate was given a measurement for every alternative, every 3 days, soon after the begin of the experiment. Before the test, the system was made to run stably for 30 min at a rated pressure first, and then the rain gauge cylinder was placed directly below the measuring point every 5 s; 12 min later, the rain tanks were removed based on placing

order and time interval of the rain tanks, and the amount of water in each rain tank was measured. In order to reduce the error of the test results, each test was repeated 3 times.

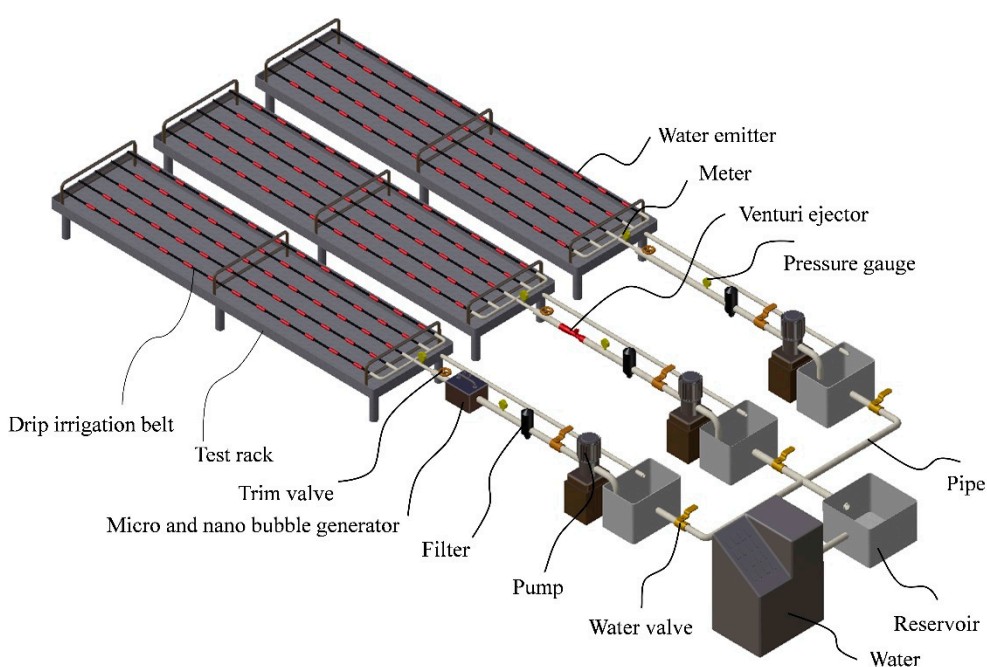

**Figure 1.** Schematic diagram of test device.

The test was run for 10 h (8:00~18:00) on a daily basis with aggregate running time of 330 h. During the test, in order to offset the water loss by evaporation and water droplets splashing, groundwater was added to the tank and the filter was cleaned before the test began in a daily manner. While operating the test device, the pressure in operation of that system remained stable at 0.1 MPa. To ensure the accuracy of the test, the capillary failed to be washed during the whole test.

The groundwater of the test site was used as the water source for the test. Table 2 expounds the water quality.

**Table 2.** Test of water quality.

| Total Nitrogen | Total Phosphorus | COD | TDS | Calcium Ions | Magnesium Ions | Iron Ions | Manganese Ion | Bicarbonate | Carbonate | Sodium Ions | pH Value | Total Salt Content | Sulfate |
|---|---|---|---|---|---|---|---|---|---|---|---|---|---|
| mg L$^{-1}$ | mg L$^{-1}$ | mg L$^{-1}$ | mg L$^{-1}$ | mg L$^{-1}$ | mg L$^{-1}$ | mg L$^{-1}$ | mg L$^{-1}$ | mg L$^{-1}$ | mg L$^{-1}$ | mg L$^{-1}$ | | S cm$^{-1}$ | mg L$^{-1}$ |
| 6.3 | 0.36 | <15 | 2.69 | 96.39 | 95.4 | 0.074 | 0.022 | 153.52 | 26.96 | 109.50 | 7.26 | 2.66 | 87.31 |

### 2.3. Research Methods

### 2.3.1. Monitoring of the Clogging Degree of the Emitter

The average discharge ratio variation Dra (%) was used in the experiment to express the clogging degree of the emitter:

$$\text{Dra}(\%) = \frac{\sum_{i=1}^{n} q_i}{n q_{new}}$$

where, $q_i$ is the flow rate of the $i$th emitter in blocking test, L h$^{-1}$;
$\bar{q}_{new}$: mean flow rate of the emitter before the test, L h$^{-1}$;
$n$: number of measured emitters.

The ratio of average emitter flow to Dra reflects the degree of reduction of average emitter flow, and a smaller Dra indicates a greater attenuation of average emitter flow and

a more serious blockage. It is universally thought that as Dra $\leq$ 75%, that emitter is blocked (ISO/TC 23/SC 18) [28]. Therefore, this test uses the average flow ratio Dra $\leq$ 75% as the judgment basis for emitter blockage, and defines varied levels of emitter blockage based on the size of Dra: When the emitter Dra $\geq$ 95%, it is defined as unblocked; providing 75% $\leq$ Dra < 95%, it is defined as slight blockage; as 50% $\leq$ Dra < 75%, it is interpreted as blockage; Assuming 20% $\leq$ Dra < 50%, it is defined as serious blockage; as Dra < 20%, it is defined as thorough blockage.

2.3.2. System Performance Evaluation

Christiansen uniformity coefficient $C_u$ and statistical uniformity coefficient $U_s$ were adopted for evaluation of influence of irrigation blockage on the performance of drip irrigation system:

(1)    Christiansen uniformity coefficient

$$C_u(\%) = \left(1 - \frac{\sum_{i=1}^{N}|x_i - \overline{x}|}{\sum_{i=1}^{N}x_i}\right)$$

In the equation,
$C_u$ : Christiansen uniformity coefficient, %;
$x_i$: Observed value of water output of the $i$th emitter, mL;
$\overline{x}$: Sample mean, mL;
$N$: indicates the number of measurement points.

(2)    Statistical uniformity coefficient

$$U_s(\%) = (1 - S/\overline{x})$$

In the equation,
$U_s$ denotes the statistical uniformity coefficient, %;
$S$ refers to the standard deviation of the sample observation value.

(3)    System performance evaluation index

According to the ASAE standard EP 458 [29], the system performance was evaluated as "excellent" at 95%~100%. When the value stands at 60% and below, the evaluation system performance proves to be unqualified. Table 3 lists the specific evaluation indicators.

**Table 3.** Uniformity evaluation.

| $U_s$ | 100~95 | 90~85 | 80~75 | 70~65 | <60 |
|---|---|---|---|---|---|
| Grades | Excellent | Good | Medium | Inferior | Unqualified |

2.3.3. Analysis of Microbial Diversity of Emitter Plugs

The clogging mechanism of aerated drip irrigation emitter was studied from the biological perspective. At the end of the experiment, the microbial diversity of blockage substances in three drip irrigation systems was analyzed, and three samples were selected from each experimental treatment for microbial diversity statistics. Genomic DNA was extracted by DNA extraction kit, and then the concentration of DNA was detected by agarose gel electrophoresis and NanoDrop2000.Genomic DNA was used as a template to identify microbial diversity.

After preprocessing the microbial DNA test data for generation of high-quality sequences, Vsearch [30] software was adopted for classification of sequences for multiple operational taxonomic units (OTUs) based on sequence similarity. OTU classification is an artificially defined taxon. In population genetic studies, the same mark is artificially assigned to a taxon for the convenience of analysis. To understand the number of bacteria and genera in the sequencing results of a sample, it is necessary to cluster the sequences.

Through the classification operation, sequences are divided into multiple groups according to their similarity, and a group is used as an OTU. A measurement sequence similarity of 97% or greater is usually interpreted as an OTU unit.

The QIIME [31] software package was adopted for selection of the representative sequences of each OTU, and every representative sequence was compared with the database for annotation, so as to obtain the microbial diversity results of each sample. By comparing and analyzing the microbial diversity results of each drip irrigation system with the clogging rule and uniformity of the irrigator, the biological response law of blockage and air-filled drip irrigation can be reflected, and then the blockage mechanism of air-filled drip irrigation can be analyzed.

## 3. Results and Analysis

### 3.1. Influence of Air Filling on the Filling Law of the Emitter

(1)   Influence of air filling on average flow ratio Dra of the emitter

Figure 2 shows the variation in the average discharge ratio Dra of the emitter with test time under different test treatments. As can be seen from the figure, Dra decreases gradually with the increase in test time under different gas filling modes. However, under MAI, the decreasing rate of Dra in the emitter was significantly lower than that in other experimental groups, and the decreasing trend of VAI and UVI was basically the same throughout the experiment period, except that in the early stage of irrigation, the Dra of VAI was slightly higher than that of UVI.

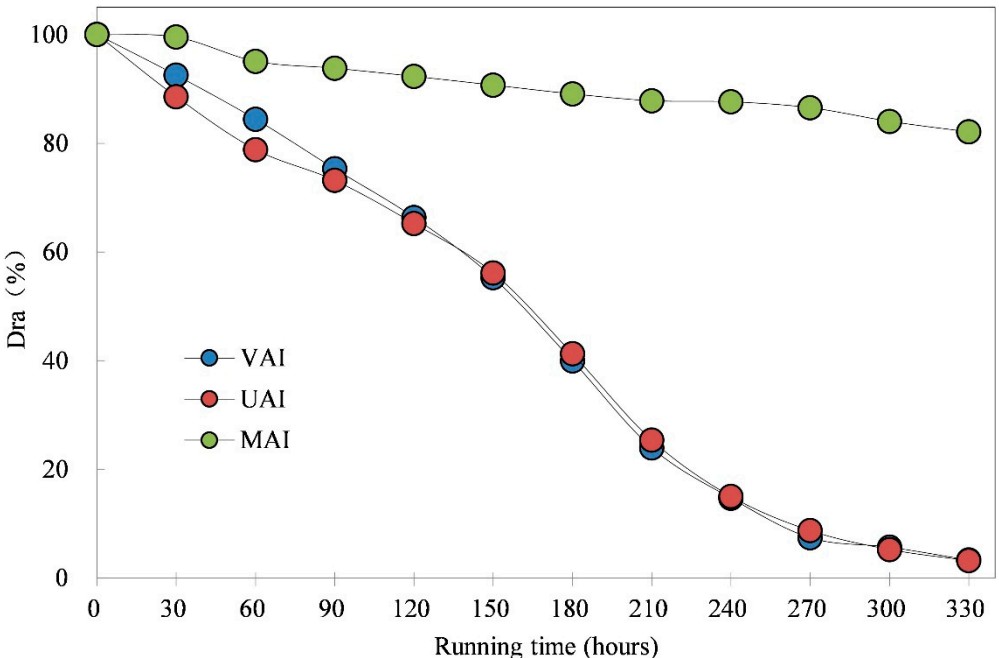

**Figure 2.** Change in average discharge ratio Dra of emitter with irrigation time.

After 90 h of work, the Dra of VAI and UVI both decreased to about 75%. After that, the VAI emitter was basically the same as that of UVI until the end of the experiment, indicating that VAI would slow down the rate of the emitter Dra decline in the early stage of irrigation.

Figure 3 shows the bar chart of the number of emitters with varied plugging degrees varies with the test time. It can be seen from the figure that in each experimental group, there was a great difference in the degree of emitter blockage under different filling modes. Among them, the time of blockage of the emitters in MAI was the latest, which was about 270 h of system operation. At the end of the test, only three emitters reached the blockage standard, and no serious blockage occurred in all the emitters.

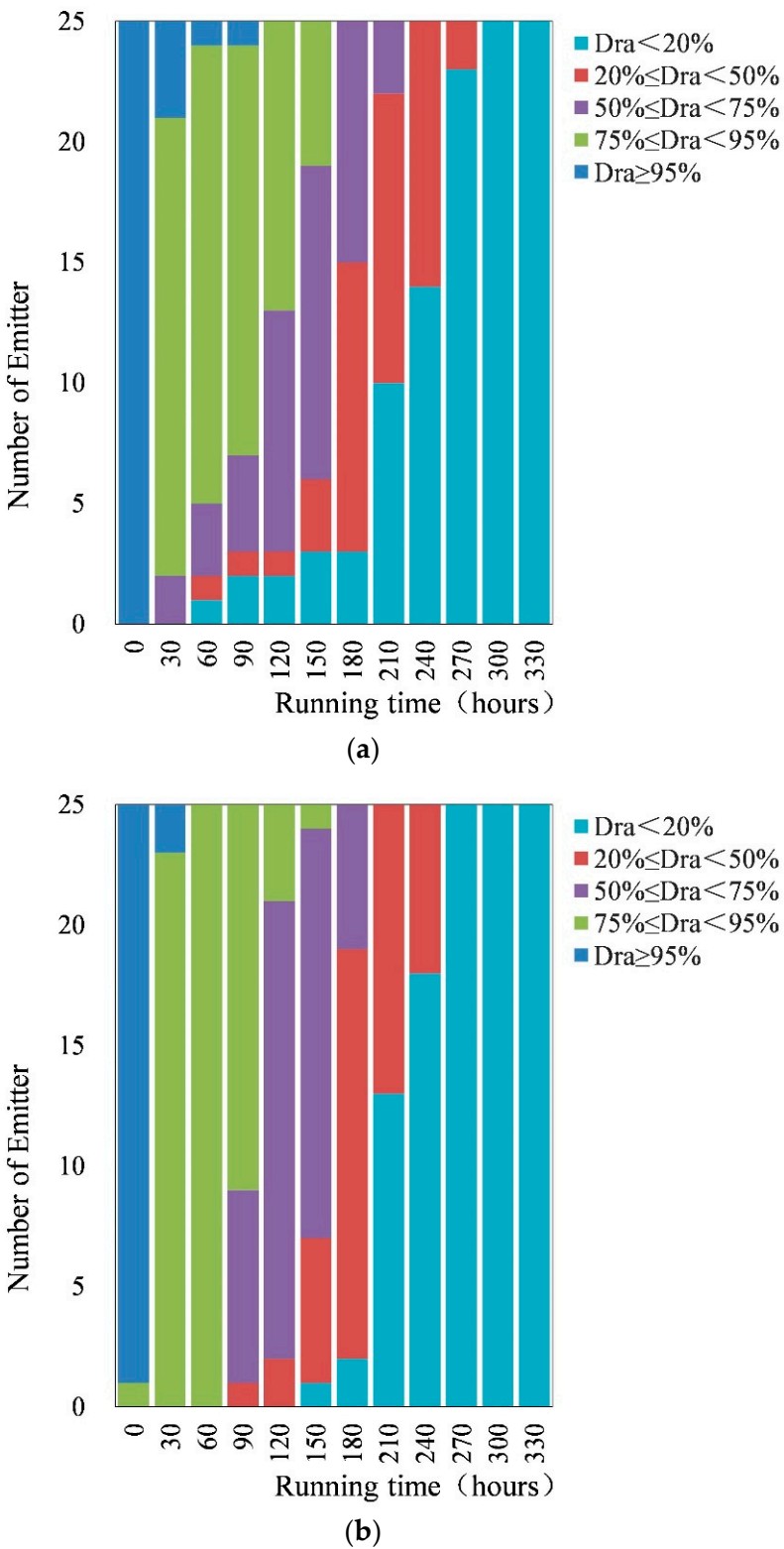

**Figure 3.** *Cont.*

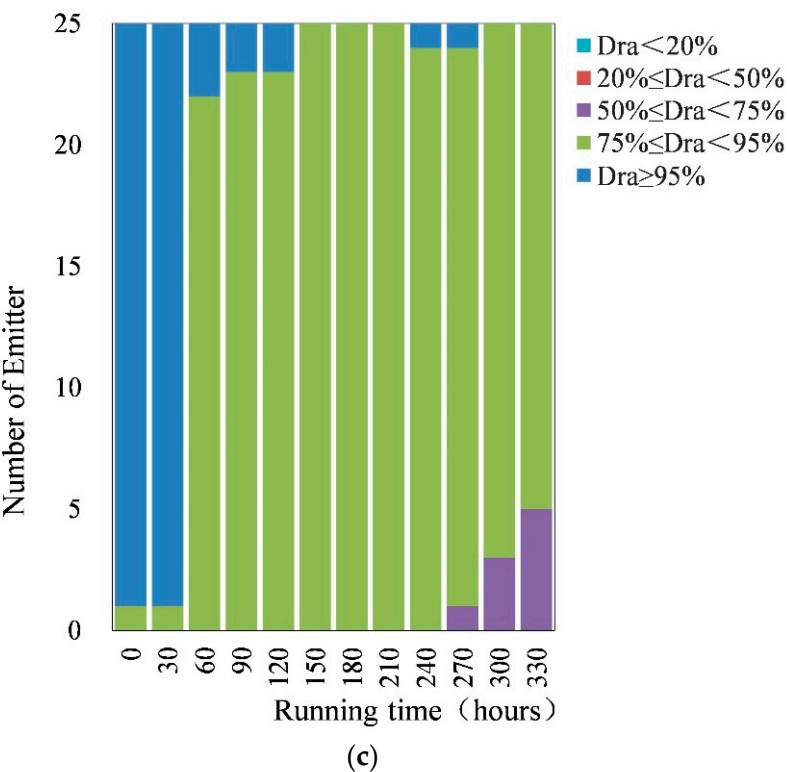

(c)

**Figure 3.** Bar chart of the number of emitters with different clogging degrees varies with irrigation time. (**a**) UAI; (**b**) VAI; (**c**) MAI.

The VAI seriously blocked the emitters after about 90 h of operation, and then the number of blocked emitters was always higher than that of other experimental groups until all the emitters were blocked. When all the emitters were seriously blocked, the system only ran for about 210 h. In UVI, serious blockage of the emitter occurred after about 60 h of operation, which was the fastest in all the tests. After about 240 h of the test, all the tested emitters in UVI reached the serious blockage standard.

(2)    The influence of air filling on the clogged space distribution of the emitter

In order to describe the dynamic development process and spatial distribution principle of emitter blockage, the dynamic change in emitter blockage at varied positions at the direction of drip irrigation belt was studied, and the dynamic change heat map of average flow ratio of emitter at different positions was drawn, as demonstrated in Figure 4. In the figure, the abscissa signifies the running time of the test, and the ordinate represents the average flow ratio Dra of the emitters at different positions at the direction of the drip irrigation belt. The size of the emitters Dra is distinguished by different colors.

As can be seen from Figure 4, as continuous advances of the test, the number of blocked emitters in each experimental group increased continuously. The blockage duration of the emitters in aerated test group was later than that in UVI, and the number of clogged emitters was also obviously smaller than that in UVI in the corresponding time. By comparison of the positions of the blocked emitters on the drip irrigation belt in each span period, it is not difficult to find that the blocked emitters in that air filling test group have a more uniform distribution on the measured drip irrigation belt.

In the direction of water flow, the drip irrigation belt is divided into three sections according to the laying length, namely, the front, middle and back sections. The proportion of the clogging emitters in this section is defined as the ratio between the quantity of clogging emitters in each section and the aggregate quantity of clogging emitters in the root drip irrigation belt. Figure 5 shows the distribution in the percentage of clogged emitters along the drip irrigation belt in each air filling mode.

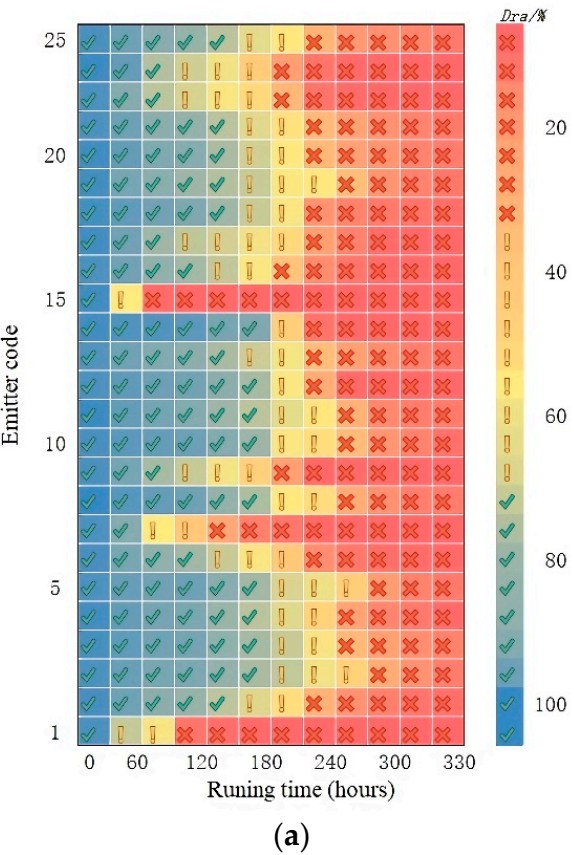

(**a**)

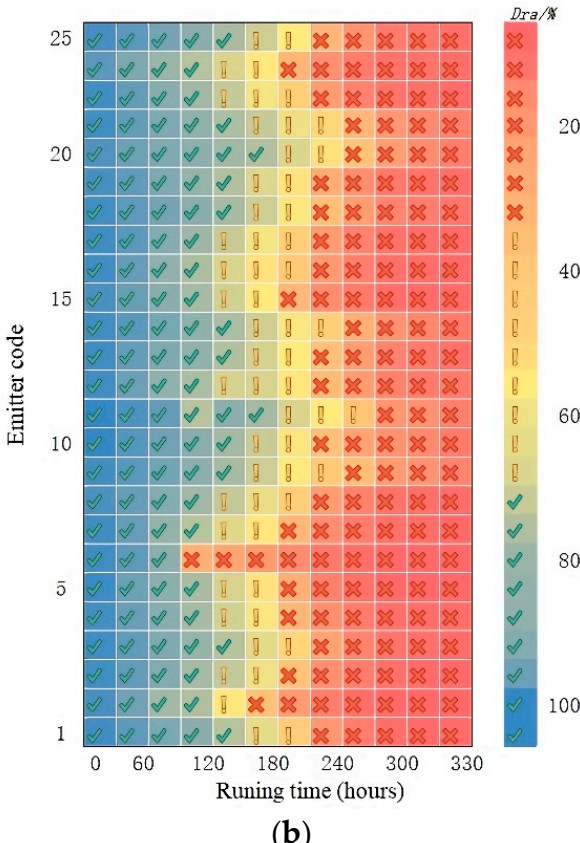

(**b**)

**Figure 4.** *Cont.*

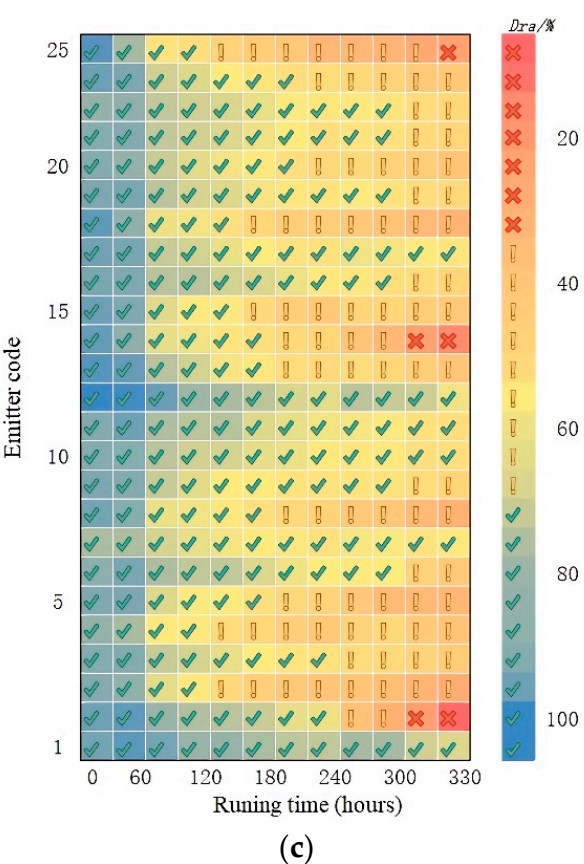

**(c)**

**Figure 4.** Distribution diagram of the emitter Dra with the change in filling time under different filling modes. (**a**) UAI; (**b**) VAI; (**c**) MAI.

It can be seen from Figure 5 that different filling modes have different influences on the change of the filling degree of the emitter, and the distribution of the clogging emitter along the drip irrigation belt is also different, which changes with the change in the filling treatment mode and the test time. In UVI, there were more clogged emitters in the front and middle sections of the drip irrigation belt, which further verified that the front section of the drip irrigation belt was more prone to blockage under normal conditions. For the gas test group, in the initial stage of blockage, blockage mainly appeared in the middle and back section. Compared with UVI, the distribution of VAI is more uniform, and the front segment blockage ratio of MAI is smaller.

The above results indicate that micro and nano inflating not only delays the time of the emitter clogging, and it also affects the spatial distribution of the emitter clogging, indicating that micro and nano inflating has an important influence on the dynamic change in the degree of emitter clogging, and makes the mechanism of emitter clogging more complex.

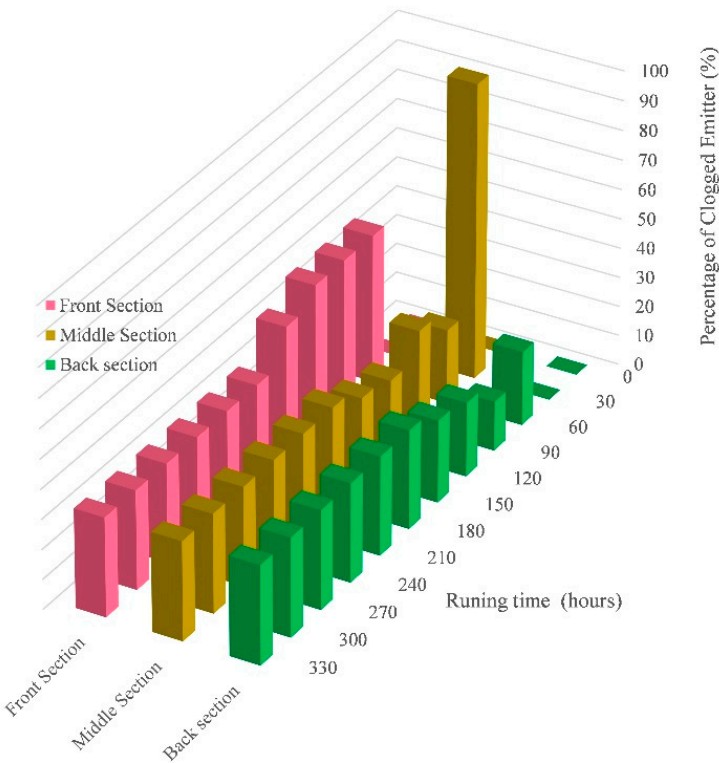

(**a**)

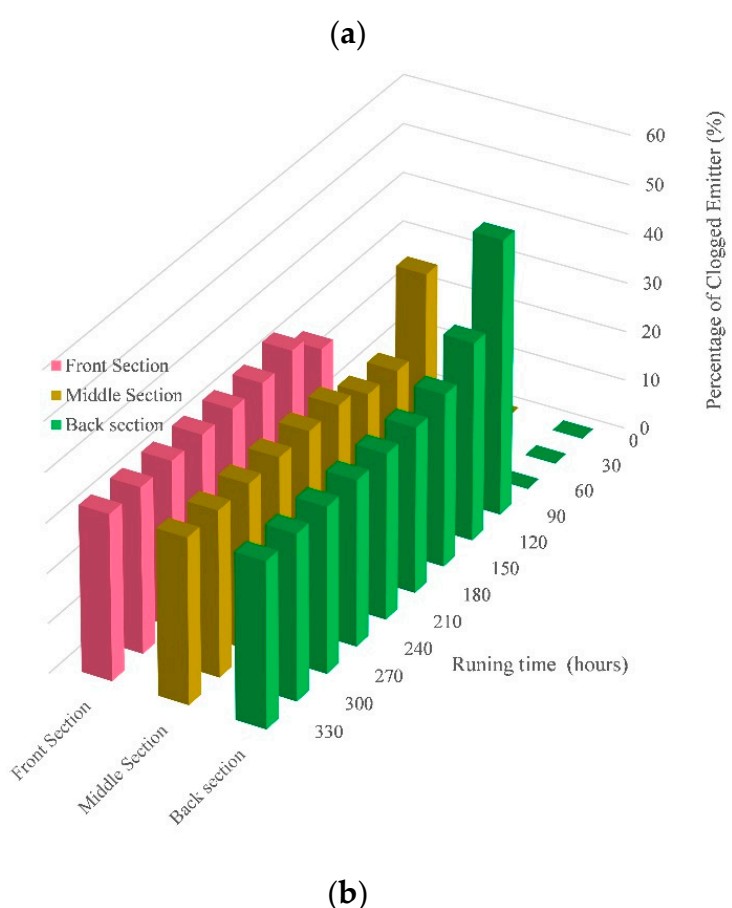

(**b**)

**Figure 5.** *Cont.*

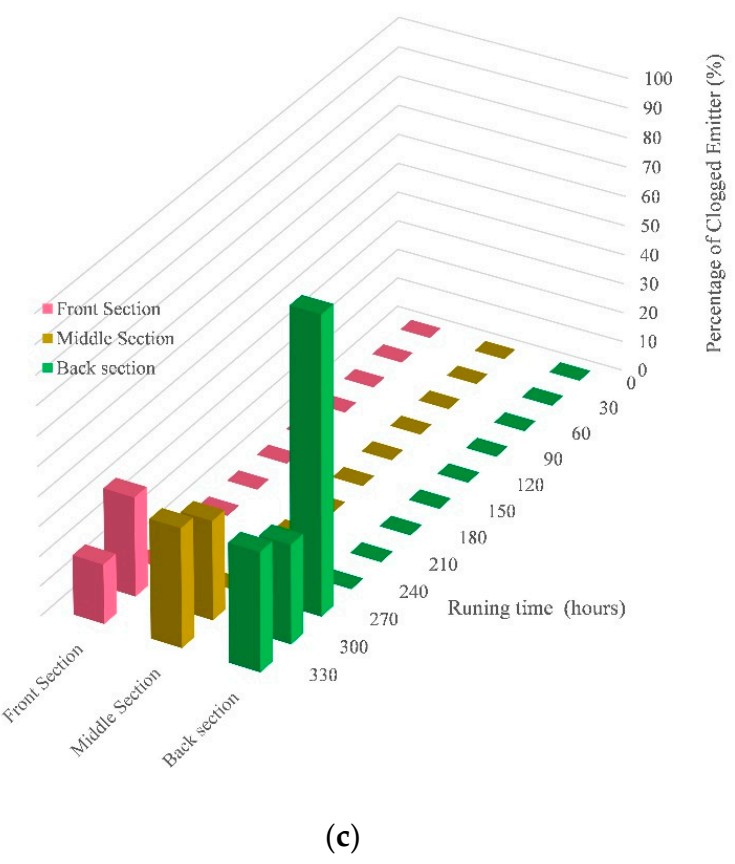

(**c**)

**Figure 5.** Distribution in the percentage of clogged emitters along the drip irrigation belt. (**a**) UAI; (**b**) VAI; (**c**) MAI.

### 3.2. Dra Regression Analysis of Emitters under Different Filling Modes

In order to analyze the statistical relationship in a quantitative manner between the average flow ratio Dra of emitter under different filling modes and the test duration (T), linear regression analysis of relevant factors was carried out in the data analysis software SPSS 22.0, and regression equations under corresponding filling modes were obtained:

Control group: Dra UA = 1.018 − 0.00330T ($R^2$ = 0.977, $p$ < 0.001);

Venturi gas test group: Dra VA = 0.992 − 0.00318T ($R^2$ = 0.980, $p$ < 0.001);

Micro and nano gas test group: Dra MA = 0.992 − 0.00052T ($R^2$ = 0.971, $p$ < 0.001);

Among them, the coefficient of determination R2 of all experimental groups was greater than 0.97, indicating a good degree of fitting of the regression equation data.

According to the regression equation, when Dra = 75%, the TUA of unaerated mode (UVI) is ≈81.21 h, the TVA of Venturi aerating mode (VAI) is ≈76.10 h, and the TMA of micro/nano aerating mode (MAI) is ≈465.38 h. TVA ≈ 0.94TUA and TMA ≈ 5.73TUA were obtained. In other words, under the same conditions, the blockage time of MAI is 5.73 times longer than that of UVI, while the blockage time of VAI is 0.94 times longer than that of UVI. Therefore, MAI can effectively delay the clogging time of the drip irrigation system, while Ventura gas (VAI) cannot.

### 3.3. Influence of Air-Fed Drip Irrigation Emitter Clogging on System Uniformity

3.3.1. Variation Law of Uniformity Coefficient of Air-Fed Drip Irrigation System

In order to study the influence of the clog of the air-fed drip irrigation emitter on the uniformity of the system, the variation in the Christiansen uniformity coefficient $C_u$ and statistical uniformity coefficient $U_s$ in different experimental groups with the running time was studied, as shown in Figure 6. It can be seen from the figure that the change

process of Christiansen uniformity coefficient $C_u$ and statistical uniformity coefficient $U_s$ with working time is similar to the change law of average flow ratio Dra.

In general, the uniformity coefficient ($C_u$ and $U_s$) of the emitter decreased with the increase in the test time under each filling mode. The Christiansen uniformity coefficient $C_u$ of VAI and MAI is better than that of UVI. The statistical uniformity coefficient ($U_s$) of Ventura gas method (VAI) is the worst, and the statistical uniformity coefficient ($U_s$) of MAI proved to be with the best performance. MAI has better uniformity coefficient stability.

According to Figure 6, the statistical uniformity coefficient $U_s$ of the emitters under the MAI and VAI modes is better than that under the UVI mode. In particular, the system uniformity of MAI was qualified throughout the whole test period, and was excellent for most of the time (about three quarters of the whole test period). At this time, the $U_s$ of UVI is reduced to approximately 40%, with obvious unqualified system performance, which is not suitable for continued use.

These results indicate that MAI can maintain good uniformity of irrigation and prolong the service span of drip irrigation systems. In the whole test process, the VAI can be evaluated as qualified for half of the time and be excellent for a quarter of the time during the test period. The UVI was qualified only a quarter of the time and excellent one-sixth of the time. It can be seen that the two air feeding methods can have certain influence on the irrigation uniformity of drip irrigation system.

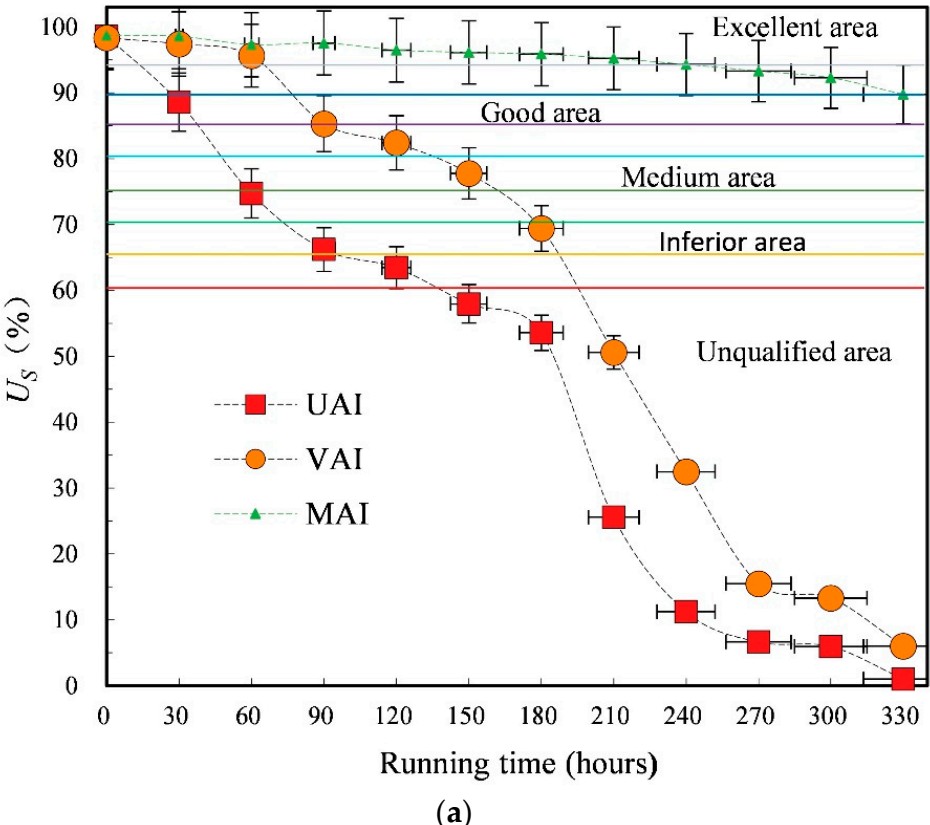

(**a**)

**Figure 6.** *Cont.*

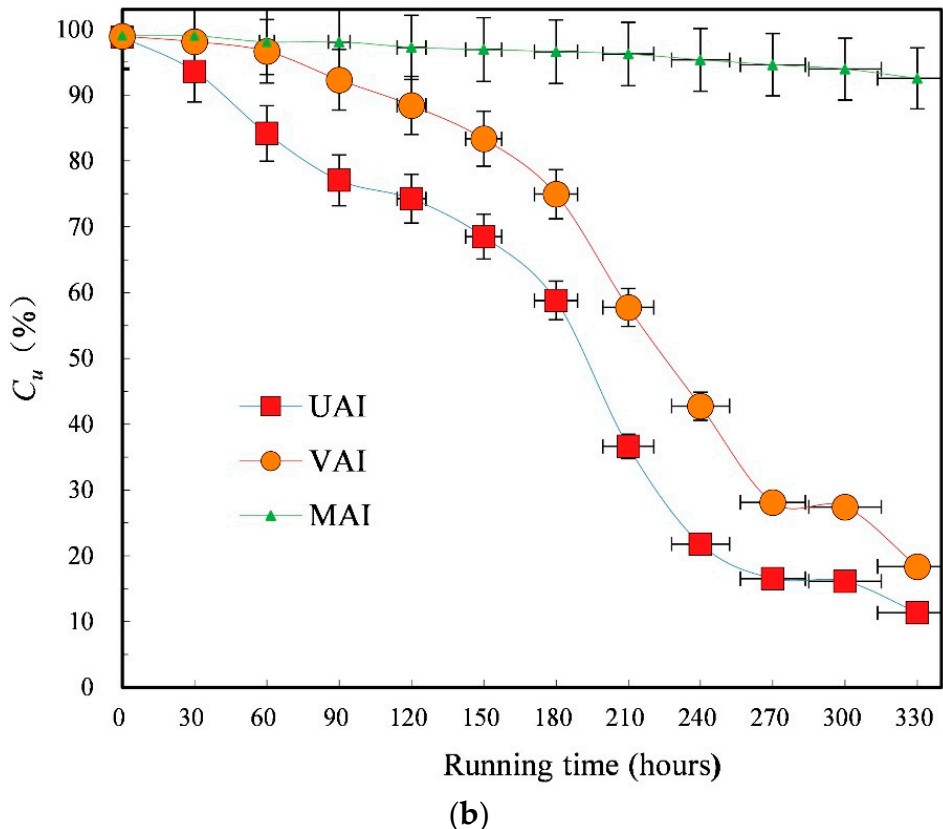

**(b)**

**Figure 6.** Variation curve of uniformity coefficient of drip irrigation system with irrigation time. (**a**) Statistical uniformity coefficient $U_s$ of drip irrigation system. (**b**) Christiansen uniformity coefficient $C_u$ of drip irrigation system.

3.3.2. Relationship between Uniformity Coefficient of Air-Fed Drip Irrigation System and Average Flow Ratio of Emitter

Figure 7 shows the relationship between the Christiansen uniformity coefficient $C_u$ and statistical uniformity coefficient $U_s$ and the average flow ratio Dra in drip irrigation system. As can be seen from the Figure, there is a good linear relationship between Dra, $C_u$ and $U_s$ under all treatments. Under the condition that the emitter Dra is the same, the effects of different filling methods on $C_u$ and $U_s$ show obvious differences.

On the whole, $C_u$ and $U_s$ under the two gas methods are better than those under the UVI. Especially when Dra is about 90%, $C_u$ and $U_s$ appear an inflection point, and with the decrease in Dra, the positive effect of MAI on $C_u$ and $U_s$ becomes more obvious. MAI reduces the sensitivity of $C_u$ and $U_s$ to Dra changes in drip irrigation systems, and makes the degree of irrigation clogging more uniform.

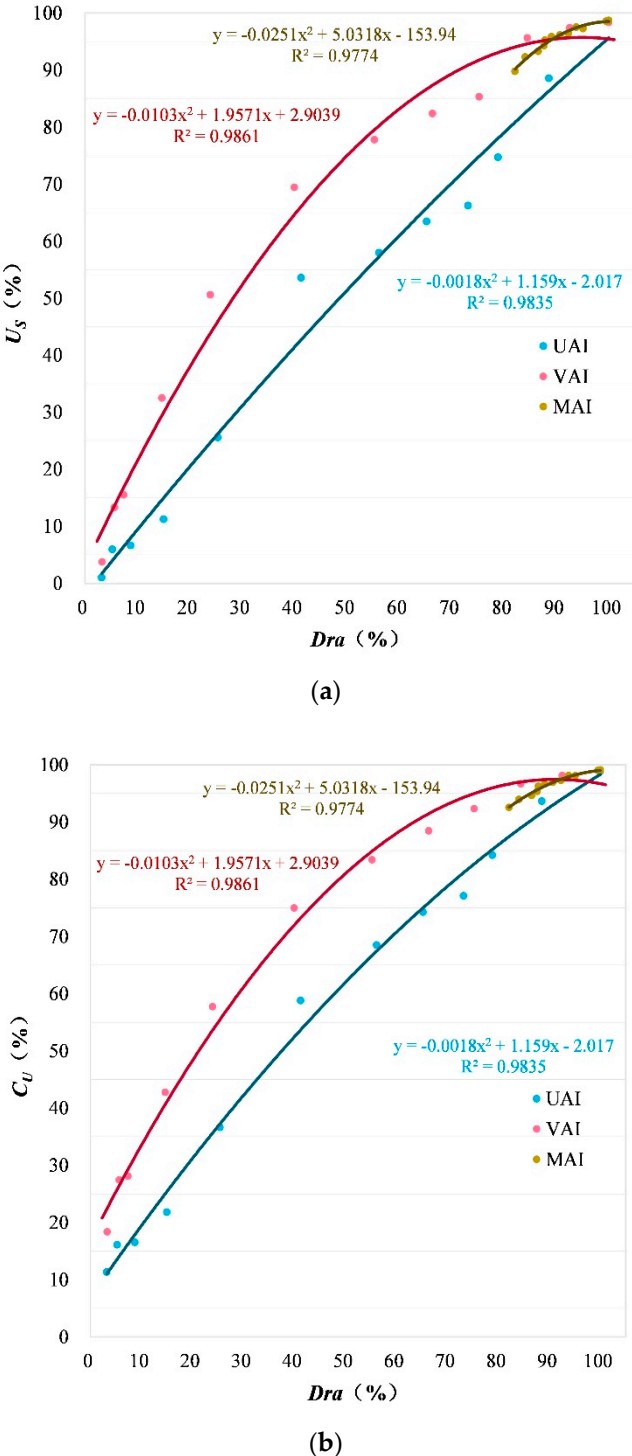

**Figure 7.** Relationship between uniformity coefficient and average flow ratio. (**a**) The relevance between the statistical uniformity coefficient $U_s$ and the mean flow ratio Dra. (**b**) The relevance between Christiansen uniformity coefficient $C_u$ and mean flow ratio Dra.

### 3.4. Microbial Diversity of Plugging Material in Air Drip Irrigation Emitter

3.4.1. Influence of Gas Addition on Microbial Quantity

After the quality control of nine samples, the microbial OTU quantity of each sample was distributed between 2109 and 2333.

First, the total number of OTUs was analyzed, and the OTU results of nine samples are shown in Figure 8.

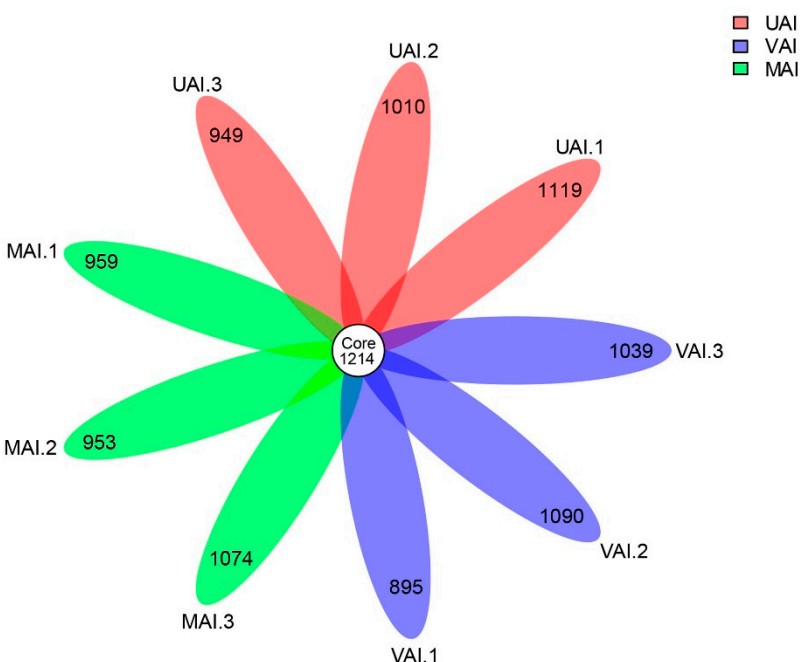

**Figure 8.** Petal diagram of OTUs distribution in each sample.

The numbers in Core represent the number of OTUs shared by the nine samples (i.e., Core OTUs).

The numbers on the petals represent the total OTUs of each sample minus the number of OTUs shared.

Based on the OTU results of every sample, the total quantity of OTUs of the nine samples was greater than the number of their own OTUs, that is, the number of the same microorganism was more than 50% of the total number of microorganisms in each sample. In order to further analyze the microbial distribution law of drip irrigation system under each gas filling method, three samples were evenly selected for each gas filling method with analysis on the average results. The microbial OTU number distribution of the stopper in the emitter under each gas filling method is shown in Figure 9.

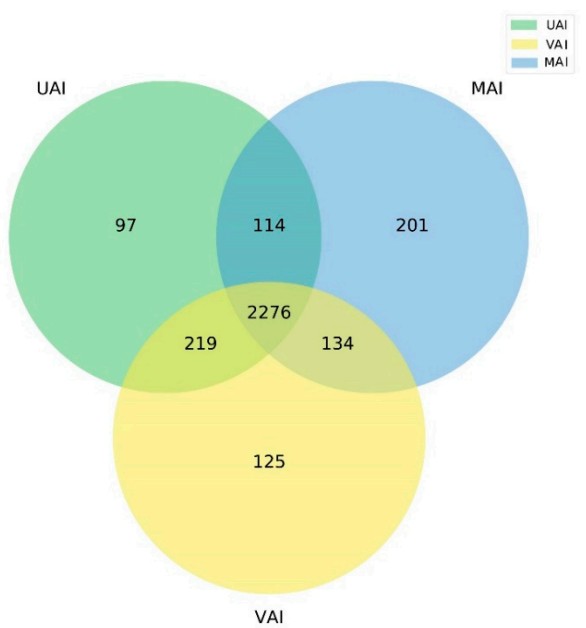

**Figure 9.** Distribution of OTUs in drip irrigation systems.

As shown in Figure 9, the total number of microbial OTUs of the emitter plugs under the three gas filling methods totaled 3166, and the total number of OTUs shared by each gas filling method stood at 2276. Among them, the number of unique OTUs in UVI was 97, the number of unique OTUs in VAI was 125, and the number of unique OTUs in the micro and MAI reached 201.The number of unique OTUs of VAI is 1.3 times that of UVI, and the number of unique OTUs of MAI is 2.1 times that of UVI. The results showed that the number of OTU of plugging material could be increased by gas injection, that is, the microbial abundance of plugging material in the emitter could be increased.

### 3.4.2. Effects of Aeration on Microbial Community Diversity

To further explore the effect of aeration on the microbial diversity in drip irrigation systems, the microbial community structure of the blockage substances in the emitter under each aeration method was analyzed by us. The number of microbial communities contained in the plugging material of the emitter under each gas filling method was counted, and the communities with the top 30 microbial numbers were analyzed. Among them, the common community species was 29, and the other unique microbial communities were represented as other. Figure 10 shows the relative abundance of microbial communities of the clogging materials in the emitters under the three gas filling methods.

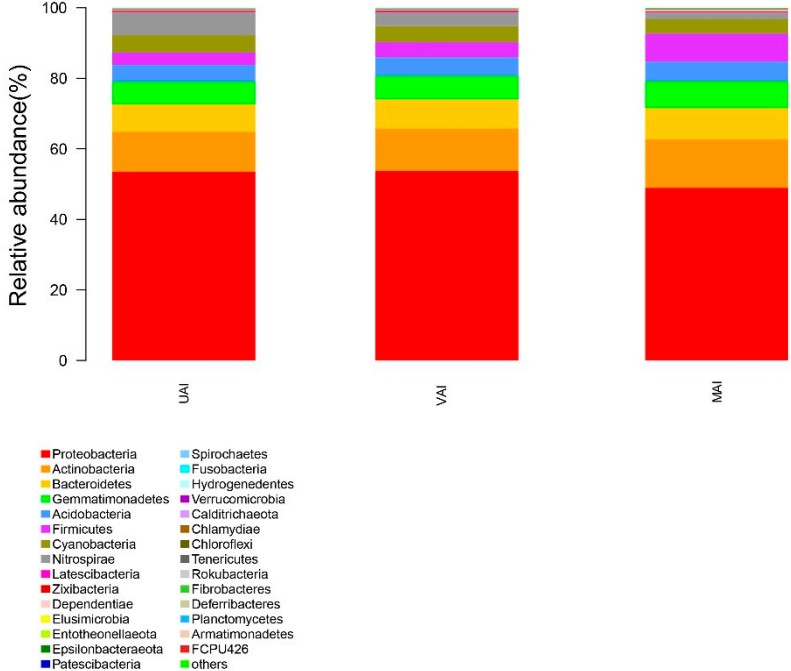

**Figure 10.** Bar chart of community structure in drip irrigation system (microbial communities with the top 30 microorganisms in the sample).

As shown in Figure 10, the relative abundance of microorganisms in VAI and UVI demonstrates similarity, and the proportion of each microbial community in the total microbial community is roughly the same, while the abundance of microorganisms in MAI shows significant difference. Among them, Nitrospirae showed significant differences among the three aeration methods, with the highest abundance in UVI and the lowest abundance in the MAI, indicating that aeration would affect the growth of microbial communities of Nitrospirae in the blocked substances of the emitter.

The abundance of Firmicutes and Actinobacteria in UVI and VAI was similar, but the abundance of the microbial community in the micro and MAI was significantly increased. The abundance of Proteobacteria in all drip irrigation systems is close to 50%, indicating that Proteobacteria is the main microorganism in the blocking substances of the irrigation system. The microbial community abundance in UVI and VAI was similar, but the mi-

crobial community abundance in MAI was significantly reduced. In conclusion, aerated drip irrigation can inhibit the growth of the Nitrospirae and Proteobacteria community structure, and promote the growth of Firmicutes and Actinobacteria microbial communities. Nitrospirae is Gram-positive bacteria, while Firmicutes, Proteobacteria and Actinobacteria are Gram-negative bacteria.

## 4. Discussion

### 4.1. Influence of Air Feeding on the Law of Emitter Blockage of Drip Irrigation System

The performance of the emitter serves as a significant factor among others that affect the overall performance of the drip irrigation system [32]. The experimental outcome demonstrates that the mean change of flow ratio (Dra) and Christiansen uniformity coefficient ($C_u$) of drip irrigation system under the three feeding methods gradually decreased with the increase in the experimental time, which was consistent with the research results of Zhou Bo [33] and Li [21]. The decreasing trend of Dra of VAI and UVI is basically the same in the whole experiment period, which is higher than that of UVI only in the early stage of irrigation. However, the Dra of VAI drops the fastest, and the whole system proves to be the first with serious blockage, and the blockage in the front middle of the system is more serious. Different from the other two gas filling methods, the decrease rate of Dra of MAI is significantly smaller. When UVI and VAI have been blocked or even completely blocked, MAI is only slightly blocked.

The Dra regression analysis of different filling modes of the emitters shows that, under the same conditions, the blockage time of MAI is 5.73 times that of UVI, and the blockage time of VAI is 0.94 times that of UVI. Therefore, MAI can effectively delay the blockage time of the drip irrigation system, while VAI cannot delay the blockage time of the emitter, which is the result of the fact that Ventura gas is sucked into the circulating water by means of pressure difference, and the distribution of gas in the water is very uneven [34].

At the same time, the bubbles produced by Ventura gas have a large diameter, which is very unstable in the system and easy to precipitate out of the water. They have a short duration in the water and cannot have an effective impact on the system performance [35]. Micro and nano bubbles, as tiny bubbles with diameters ranging from 200 nm to 50 μm, have the characteristics of super stability and strong oxidation [36]. After entering the drip irrigation system, micro and nano bubbles can produce oxidative degradation of substances that mainly cause biological clogs, such as microbial extracellular polymers on the emitter, and mitigate the impact of biological clogs on the performance of the emitter [37].

When the internal pressure is higher in the micro/nano bubbles, the bubbles burst from the relative energy. When in contact with organic matter in water, part of the organic molecular chain structure will break, hindering the $SiO_2$ and $Al_2O_3$, and the physical blocking formation has a strong adsorption of particles together relieving the physical plugging effect of the performance of the irrigation device [38]. Studies have shown that, similar to nanoparticles, micro and nano bubbles carry significant surface negative charges, which will repulse with negatively charged sulfate and phosphate oxygen-containing groups, hindering their binding with calcium and magnesium ions, reducing the formation of inorganic solids in the drip irrigation system, and slowing down the impact of chemical blockage on the drip irrigation system. In addition, micro and nano bubbles can change the hydrodynamic characteristics in the drip irrigation system and reduce the migration resistance of impurities that can be caused by wall pairs, so as to effectively improve the migration efficiency of impurities and slow down the settlement of impurities [39,40]. Therefore, MAI can significantly change the filling process of the emitter and effectively alleviate the clogging of drip irrigation systems.

### 4.2. Influence of Air Feeding on Uniformity of Drip Irrigation System

The results of uniformity analysis of drip irrigation system showed that the uniformity of the system under MAI was qualified during the whole experimental period, and the uniformity was excellent for most of the time (about 3/4 of the whole experimental period).

Under VAI, only 1/2 of the time during the test period can be evaluated as qualified, and 1/4 of the time can be evaluated as excellent. Under UVI, only 1/4 of the time is qualified, and 1/6 of the time is excellent. It can be seen that different feeding methods have certain influence on the uniformity of irrigation water in the whole working process of drip irrigation system. Among them, MAI is obviously beneficial to maintain good uniformity of drip irrigation systems, so as to prolong the service life of the drip irrigation system.

It is the result that the Venturi bubbles under VAI are only mm in diameter and have a small specific surface area, which cannot effectively adsorb tiny impurities in water and affect the uniformity of the distribution of blocking substances in the irrigation system. However, the diameter of micro/nano bubbles under the MAI method is μm. When the volume is equal, the specific surface area of the micro/nano bubbles with a radius of 10 μm is 100 times that of the Venturi bubbles with a radius of 1 mm [41]. Therefore, compared with VAI, the bubbles produced by MAI have stronger adsorption ability, which can effectively adsorb the tiny impurities in the irrigation water and drive the tiny impurities to migrate in the water.

In addition, compared with the principle of pressure difference inspiration of Venturi gas method (VAI), MAI is to inhale and break air into bubbles and distribute them in water with relatively uniform distribution [42], thus making the distribution of blocking substances in the system more uniform. At the same time, micro and nano bubbles can produce a strong long-term hydrophobic force in the bridging process, which can improve the adhesion ability of impurities and bubbles [43,44], reduce the probability of falling off, and promote the migration of water flow to impurities and other substances that are easy to form irrigation plugs, so as to improve the uniformity of drip irrigation system.

### 4.3. Influence of Air Refueling on Microbial Diversity of Clog in Emitter

Aeration can increase the oxygen content of water, promote the oxidation of organic matter, and affect the microbial diversity of the plug in the emitter. The results showed that both VAI and MAI could increase the number of microbial communities and change the activity of some microorganisms. Through analysis on microbial diversity, it was found that MAI inhibited the growth of Gram-positive bacterial community structure and prevented it from producing biofilms (microbial biofilm), which are not easily degradable and hinder/block the emitters easily. In addition, MAI promoted the growth of Gram-negative bacterial community, while VAI did not.

This is because the micro and nano bubbles produced by MAI carry significant surface negative ions, which move slowly in water, and can fully oxidize microorganisms when the bubbles burst [23]. However, the bubbles generated by VAI are prone to bubble aggregation with inconsistent gas output, and the oxygen dissolution efficiency is at a low level, which failed to produce enough oxidation effect on microorganisms [45,46]. At the same time, the cell membrane structure of Gram-negative bacteria is more complex than that of Gram-positive bacteria [47], and the energy generated when micro and nano bubbles burst is not enough to damage the cell membrane structure.

According to the characteristics of Gram-positive and Gram-negative bacteria, it is difficult for Gram-negative bacteria to secrete protein [48,49], while it is an easy task for Gram-positive bacteria to achieve it, so their non-cell wall structure peptidoglycan content is at a high level. The oxidation of non-cell wall structures of Gram-positive bacteria by micro and nano bubbles can damage them and inhibit their growth. Ushida A et al. [50] also found that the extracellular polymer content on the surface of nylon membrane could be reduced when treated with micro and nano bubble water.

At the same time, the extracellular polymer has a certain viscosity, which can agglomerate microorganisms and water impurities, and aggravate the clogging of drip irrigation system [51]. This indicates that micro and nano bubbles can reduce the content of peptidoglycan in the biofilm microbial community to reduce the content of extracellular polymer in biofilm, and then slow down the impurities and biofilm aggregation in the irrigation system,

and finally improve the anti-blockage performance of the irrigation system. Therefore, MAI can be used as a drip irrigation method to relieve the clogging of a drip irrigation system.

In this paper, the law of emitter blockage, system uniformity and microbial diversity of blockage substances in drip irrigation systems with different feeding methods were studied. Due to the limitations of the experiment, only a single groundwater circulation test was conducted. In the next step, we will comprehensively study the effects of aeration on irrigation blockage and the microbial diversity of blockage substances under different water quality, such as reclaimed water and biogas slurry, and systematically analyze the effects of aeration drip irrigation on soil and crops.

### 5. Conclusions

The effects of aeration on the blockage of the emitter and the microbial diversity of the clogging material were evaluated through experiments. The conclusions are as follows:

1.  Inflating can affect the blockage performance of the emitter; in particular, the micro/nano inflating mode can significantly optimize the anti-clogging performance of the drip irrigation system and prolong the service life of the drip irrigation system.
2.  The nano-aerating mode can maintain good uniformity in the drip irrigation system, reduce the sensitivity of Christiansen uniformity coefficient Cu and statistical uniformity coefficient Us to the change in average flow ratio Dra, and effectively extend the service life of drip irrigation systems, while the influence of the Venturi aerating mode is limited.
3.  The micro and nano gas filling method can increase the microbial abundance of the clogging substance of the emitter, significantly inhibit the growth of the Gram-positive bacterial microbial community, promote the growth of the Gram-negative bacterial microbial community, reduce the extracellular polymer content of microorganisms and effectively alleviate the impact of microorganisms on the blockage of the emitter.

**Author Contributions:** P.L.: Conceptualization, Methodology, Formal analysis, Investigation, Data curation, Writing, Visualization, Project administration. H.L.: Conceptualization, Methodology, Validations and Formal analysis. J.L.: project administration, Formal analysis. X.H.: Reviewing, Editing and Supervision. Y.L.: Investigation and Data curation. Y.J.: Conceptualization, Methodology, Resources. All authors have read and agreed to the published version of the manuscript.

**Funding:** This research was funded by the National Natural Science Fund of China, grant number is 52009137, the Scientific and Technological Project of Henan Province, grant number is 202102110279; the Central Public-interest Scientific Institution Basal Research Fund, grant numbers are Y2021YJ07 and Y2022PT13; Science and Technology Major Project of Henan Province, grant number is 221100110700; the Basic Research Project of the Farmland Irrigation Research Institute (FIRI) of the Chinese Academy of Agricultural Sciences (CAAS), grant number is FIRI2022-10.

**Institutional Review Board Statement:** Not applicable.

**Informed Consent Statement:** Not applicable.

**Acknowledgments:** We are grateful for the National Natural Science Fund of China (52009137), the Scientific and Technological Project of Henan Province (202102110279), the Central Public-interest Scientific Institution Basal Research Fund (Y2021YJ07/Y2022PT13), Science and Technology Major Project of Henan Province (221100110700), the Basic Research Project of the Farmland Irrigation Research Institute (FIRI) of the Chinese Academy of Agricultural Sciences (CAAS) (FIRI2022-10).

**Conflicts of Interest:** The authors declare that they have no known competing financial interests or personal relationships that could have appeared to influence the work reported in this paper.

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
