# Peer review of "Effect of Aeration on Blockage Regularity and Microbial Diversity of Blockage Substance in Drip Irrigation Emitter"

_agriculture, doi:10.3390/agriculture12111941_

Round 1

Reviewer 1 Report

the unit representations I showed in the pdf file will be fixed

The graphics on pages 10 and 11 have a correction in the lineup.

The technical drawing of the emitter should be attached to the publication.

The specifications of all equipment used in the test unit should be given in detail.

Author Response

please find the responses in the attached file

Reviewer 2 Report

Reviewer Comments:

1.      Avoid repetition of words in the same sentence, and simplify the phrases in the abstract as mentioned in the comments

2.      Minor addition/deletions in the introduction part can be revised according to the comments made

3.      Blockage of emitters under UAI, VAI and MAI can also be explained in the perspective of microbial abundance over the experimental period, rather than the whole at once

4.      Abundance of microbial community cannot be held sole responsible for blockage of emitters, since the water do contain the salts which form precipitates. Water quality testing for salts at the end of the experiment would have been strengthen the findings further

5.      Minor corrections in the discussion area can be corrected according to the suggestions made in the manuscript.

6.      Usage of any drip irrigation method last for the entire cropping period/ over  along period in the plantation or horticultural crops, the experiment can be extended further for much better results

7.      Check the repetition of the numerical sequence in the references

Author Response

(The authors gave the same response as above.)
